# Perspectives on the Role of Non-Coding RNAs in the Regulation of Expression and Function of the Estrogen Receptor

**DOI:** 10.3390/cancers12082162

**Published:** 2020-08-04

**Authors:** Mohammad Taheri, Hamed Shoorei, Marcel E. Dinger, Soudeh Ghafouri-Fard

**Affiliations:** 1Urogenital Stem Cell Research Center, Shahid Beheshti University of Medical Sciences, Tehran 16666-63111, Iran; mohammad.taheri@sbmu.ac.ir; 2Department of Anatomical Sciences, Faculty of Medicine, Birjand University of Medical Sciences, Birjand 9717853577, Iran; Shoorei.h@tbzmed.ac.ir; 3School of Biotechnology and Biomolecular Sciences, University of New South Wales, Sydney, NSW 2052, Australia; 4Department of Medical Genetics, Shahid Beheshti University of Medical Sciences, Tehran 19839-63113, Iran

**Keywords:** lncRNA, miRNA, estrogen receptor, cancer

## Abstract

Estrogen receptors (ERs) comprise several nuclear and membrane-bound receptors with different tissue-specific functions. ERα and ERβ are two nuclear members of this family, whereas G protein-coupled estrogen receptor (GPER), ER-X, and Gq-coupled membrane estrogen receptor (Gq-mER) are membrane-bound G protein-coupled proteins. ERα participates in the development and function of several body organs such as the reproductive system, brain, heart and musculoskeletal systems. ERβ has a highly tissue-specific expression pattern, particularly in the female reproductive system, and exerts tumor-suppressive roles in some tissues. Recent studies have revealed functional links between both nuclear and membrane-bound ERs and non-coding RNAs. Several oncogenic lncRNAs and miRNAs have been shown to exert their effects through the modulation of the expression of ERs. Moreover, treatment with estradiol has been shown to alter the malignant behavior of cancer cells through functional axes composed of non-coding RNAs and ERs. The interaction between ERs and non-coding RNAs has functional relevance in several human pathologies associated with estrogen regulation, such as cancers, intervertebral disc degeneration, coronary heart disease and diabetes. In the current review, we summarize scientific literature on the role of miRNAs and lncRNAs on ER-associated signaling and related disorders.

## 1. Introduction

Estrogen receptors (ER) are classified into two major groups, ERα and ERβ, which are produced by *ESR1* and *ESR2* genes, respectively. ERs are activated by the estrogen hormone and have specific tissue- and cell-type expression patterns [1]. The activation of ER by estrogen leads to the translocation of ER into the nucleus and subsequent binding with target DNA sequences to alter the expression of certain genes [2]. ERα participates in the development and function of several bodily organs such as the reproductive system, brain, heart and musculoskeletal systems [3]. In addition to the female reproductive system and mammary gland, this type of ER has widespread expression across the body [3,4]. Accordingly, the functions of several organ systems are commonly affected in ERα knockout mice [4]. Most notably, a kind of ligand-independent signaling has been reported for ERα, in a manner in which this receptor can be activated by either epidermal growth factor (EGF) or insulin-like growth factor-1 [5,6]. ERβ is another member of the ER family that, following binding to 17-β-estradiol (E2), estriol or similar ligands, makes homodimers or heterodimers with ERα. These dimers bind with and activate the transcription of target genes. Notably, a dominant negative effect has been reported for some isoforms of ERβ that leads to the inhibition of the activity of other ERs [7]. Moreover, the anti-proliferative effect of ERβ can compete against the functions of ERα in the reproductive system [8]. ERβ also has a tumor-suppressive role in a variety of cancers, including prostate malignancy [9].

In addition to these nuclear receptors, a number of membrane ERs including GPER, ER-X, and Gq-mER have been recognized that are mainly G protein-coupled proteins [10]. These receptors stimulate intracellular signaling through the induction of adenylyl cyclase and G-protein-associated release of membrane-linked heparin-bound EGF [11]. These receptors have putative functions in neurons. For instance, ER-X expression has been documented in the neonate cortex. It is also re-activated in the adult brain following ischemic brain stroke [10]. The functions of membrane ERα signaling in response to estrogen are exerted in a highly tissue-dependent manner. For instance, the trabecular bone in the axial skeleton is highly dependent on this pathway, while the effects of estrogen on liver weight and total body fat mass are essentially independent of this pathway [12]. Thus, both nuclear and membrane-bound ERs have various functions in diverse human tissues. Consequently, the dysregulation of their expression can interfere with the physiological conditions of related organs. Figure 1 provides a schematic overview of the estrogen receptor pathway and its key functions.

Recent studies have revealed functional associations between different ERs and non-coding RNAs. Several long non-coding RNAs (lncRNAs) and microRNAs (miRNAs), which are abundantly expressed in mammalian genomes, affect or are influenced by ERs. Moreover, a number of studies have indicated associations between circular RNAs (circRNAs) and ERs. LncRNAs are typically defined as transcripts larger than 200 nucleotides with no recognizable protein-coding capacity and can interact with various biomolecules including RNA, DNA and proteins. Their regulatory roles in gene expression can be exerted through the modulation of chromosome configuration, the regulation of transcription, splicing, availability and stability of mRNA, and post-translational modifications [13]. miRNAs, which are ~22 nucleotides in length, mainly function as post-transcriptional regulators of gene expression through mediating the degradation and/or translational suppression of their target mRNAs. In addition, they exert specific functions in the nucleus, such as miRNA-guided transcriptional regulation of gene expression [14]. CircRNAs are mostly produced by the alternative splicing of pre-mRNA. With a specific structure formed by the binding of their 3′ and 5′ ends, they have specific functions in normal development and human disorders, several of which need to be clarified [15]. In the current review, we summarize the scientific literature published to date on the role of noncoding transcripts in ER-associated signaling.

## 2. LncRNAs and ER Functions in Cancers

ER signaling has been implicated in several aspects of tumorigenesis such as cancer initiation, progression and metastasis. ER co-regulatory molecules are differentially expressed in malignant cancers and their activities can be changed during cancer progression [16]. The ERα coregulators Amplified in breast cancer 1 (AIB1) and steroid receptor coactivator (SRC)-1 have been demonstrated to facilitate breast cancer metastasis through the induction of matrix metalloproteinase 2 (MMP2) and MMP9 and twist, respectively [17,18].

Moreover, ER signaling contributes to the stemness properties in the context of cancer. The expression of ER in breast cancer cells has been correlated with the expression levels of cancer stem cell (CSC) markers Gli1 and ALDH1. In addition, E2 has been shown to increase Gli1 expression only in ER-positive breast cancer cells. E2 has enhanced the self-renewal of CSCs as well as possessing invasive properties and epithelial–mesenchymal transition (EMT) in ER-positive/Gli1 positive cells but not in Gli1 knockdown cells. Therefore, estrogen effects on CSCs and EMT are mediated by Gli1 [19]. Moreover, a certain variant of ER, namely ERα36, has been involved in tamoxifen-induced stemness in breast cancer cells. Tamoxifen has a direct interaction with this ER variant which stimulates ERα36 to increase the stemness and metastatic potential of these cells through the activation of Aldehyde Dehydrogenase 1 Family Member A1 (ALDH1A1) [20].

ERα is among the transcription factors that induce the promoter activity of the insulin-like growth factor receptor (IGFR) gene, leading to the over-expression of IGF-1R and evasion from anti-growth signals [21]. On the other hand, estrogen has been shown to induce apoptosis in several types of cancer cells. Several signaling pathways such as the intrinsic and extrinsic apoptosis pathways, the NF-κB-associated survival pathway and PI3K/Akt axis participate in E2-associated apoptosis [22].

ERs have additional roles in the regulation of immune cell function in the tumor microenvironment. These kinds of receptors are extensively present in several cell types participating in the innate and adaptive immune reaction and modulate cytokines secretion. Investigations conducted in several cancer types, such as carcinomas, being conventionally considered as non-immunogenic cancers, indicate estrogen as a possible mediator of immunosuppression via the regulation of pro-tumor reactions in an independent manner from their effects on tumor cells [23].

Consistent with these diverse roles of ER signaling in carcinogenesis, ER-associated lncRNAs are also involved in several aspects of tumorigenesis. LncRNAs induce several crucial cancer phenotypes via their interactions with numerous biomolecules such as DNA, protein, and RNA [24]. The functions of ER-associated lncRNAs have been assessed in diverse cancers; however, breast cancer has been a particular focus due to the importance of ER signaling in breast tumorigenesis. Both types of nuclear and membrane-bound ERs have been investigated. For instance, the expression of the estrogen-inducible TMPO Antisense RNA 1 (TMPO-AS1) is increased in endocrine therapy-resistant MCF-7 cells compared with its expression in regular MCF-7 cells. This lncRNA enhances the proliferation and survival of ER-positive breast cancer cells. High-throughput studies showed associations between this transcript and the estrogen signaling pathway. Mechanistically, TMPO-AS1 regulates *ESR1* expression via direct interaction with its mRNA by increasing the stability of the transcript [25].

The activation of GPER by E2 and the GPER-specific agonist G1 GPER has been shown to decrease the expression of lncRNA-Glu [26]. This lncRNA can diminish the function of glutamate transport and the expression of VGLUT2. Thus, GPER-associated decrease in lncRNA-Glu enhances glutamate transport function and the expression of VGLUT2. LncRNA-Glu-VGLUT2 signaling boosts the effects of cAMP-PKA signaling in the production of glutamate in triple-negative breast cancer cells, therefore increasing the invasive and metastatic potential of these cells [26].

The lncRNA in non-homologous end joining pathway 1 (LINP1) is overexpressed in tamoxifen-resistant breast cancer cells [27]. In vitro and in vivo investigations showed that LINP1 silencing decreased resistance to this drug and the viability of resistant breast cancer cells. Moreover, LINP1 silencing enhances apoptosis in these cells after exposure to tamoxifen. On the other hand, LINP1 upregulation enhances cell mobility through influencing the epithelial–mesenchymal transition process. The transcription of this lncRNA is reduced by ER. Both tamoxifen therapy and hormone withdrawal upregulate LINP1. The upregulation of this lncRNA has been associated with reduction in ER levels and the estrogen response, which is an essential element in anti-estrogen resistance. Thus, LINP1 has an important role in tamoxifen resistance and could be a putative target to enhance the efficiency of tamoxifen therapy in breast cancer [27].

An 11-lncRNA signature has been shown to predict the prognosis and probability of recurrence in a cohort of ER-positive breast cancer patients who received tamoxifen [28]. Furthermore, upregulation of lncRNA LINC01116 in breast cancer tissues has been shown to correlate with patient outcome, tumor size and cancer stage. This lncRNA serves as a molecular sponge for miR-145, enhancing the expression of its target gene *ESR1*. Based on the results of clinical studies, LINC01116 has been suggested as a prognostic marker in breast cancer that influences the progression of this disorder through the modulation of *ESR1* expression [29].

The expression of the oncogenic lncRNA H19 has been recognized as a parameter that determines resistance to paclitaxel in ERα-positive breast cancer cells, but not in ERα-negative cells. This lncRNA reduces paclitaxel-induced cell apoptosis through the suppression of the expression of pro-apoptotic genes Bcl-2-interacting killer (BIK) and NOXA. Notably, H19 has been identified as a downstream target of ERα. Thus, modulation of ERα expression can influence H19 expression, thus altering paclitaxel resistance in breast cancer cells. Taken together, the ERα/H19/BIK axis has been suggested as a key factor in the determination of chemoresistance [30]. The estradiol-induced expression of H19 is also involved in the pathogenesis of papillary thyroid carcinoma. In vitro studies show that estradiol enhances H19 expression through ERβ. In papillary thyroid carcinoma cells, H19 serves as a molecular sponge for miRNA-3126-5p to mutually induce ERβ expression. Taken together, the ERβ-H19 positive feedback participates in the maintenance of cancer stem cells in this type of cancer under treatment with estradiol. This axis is suggested as a putative therapeutic target in this type of thyroid cancer [31].

The expression of the lncRNA myocardial infarction associated transcript (MIAT) is increased in ER-positive breast cancer tissues and in the MCF-7 cell line. Diethylstilbestrol has increased MIAT expression in MCF-7 cells in an ERα-dependent way. MIAT knockdown reduced diethylstilbestrol (DES)-associated MCF-7 cell proliferation. MIAT has been suggested as a putative biomarker and a possible therapeutic candidate for ER-positive breast cancer [32]. Moreover, the expression of this lncRNA has been elevated in high-grade breast tumors compared to low-grade tumors. This lncRNA was over-expressed in ER-, progesterone receptor (PR)- and Her2-positive breast cancer tissues. MIAT silencing inhibited breast cancer cell proliferation and induced cell cycle arrest. Moreover, THE silencing of MIAT enhanced apoptosis and reduced the migratory potential of these cells. Such effects were accompanied by THE over-expression of miR-302, miR-150, and THE downregulation of miR-29c [33]. Another high-throughput sequencing study in MCF-7 cells demonstrated estradiol-mediated transcriptional alterations of lncRNAs and their target genes. Moreover, the transcriptional response of lncRNAs to estradiol mostly occurred before their target genes [34]. ERα-mediated suppression of the MTA1 lncRNA has been shown to reduce the cell proliferation and metastasis of hepatocellular carcinoma [35]. ERα stimulation by estradiol enhances the proliferation of luminal breast cancer cells. In addition, ERα has additional hormone-independent activities to preserve the epithelial features of breast cancer cells. ERα silencing has altered the expression of several genes including 133 Apo-ERα-associated lncRNAs.

The most prevalent ApoER-associated lncRNA is DSCAM Antisense RNA 1 (DSCAM-AS1). This lncRNA is expressed in ERα+ breast cancer cells, but is absent in pre-neoplastic tissues. Notably, the expression of DSCAM-AS1 has been inversely correlated with EMT markers. DSCAM-AS1 silencing recapitulated the impact of ERα knockdown such as growth arrest and the activation of EMT markers [36].

The upregulation of HOX Transcript Antisense RNA (HOTAIR) in ER-positive breast cancer cells increases cell proliferation, growth and invasion and has been associated with risk of metastasis. Moreover, estradiol can bind with the estrogen response element in the promoter region of this lncRNA. Notably, HOTAIR activity has a crucial role in the induction of ER signaling in malignant cells [37]. Another study in prostate cancer cells revealed opposite functions of metastasis associated lung adenocarcinoma transcript 1 (MALAT1) and HOTAIR in estrogen-associated transcriptional modulation. Treatment of these cells with 17β-estradiol enhanced HOTAIR recruitment to chromatin but decreased the recruitment of MALAT1. Therefore, the interaction between lncRNAs, estrogens and ERs has a crucial function in the modulation of gene expression in prostate cancer [38].

LncRNAs can modulate both the expression and activity of ER. The expression of HOTAIR has been increased in tamoxifen-resistant breast cancer tissues compared to the primary samples. This lncRNA is directly targeted by ER. ER signaling has been shown to suppress the expression of HOTAIR. Notably, the over-expression of HOTAIR enhances the ER protein level and therefore increases ER’s tenancy on the chromatin. This type of interaction facilitates the regulation of downstream genes by ER. HOTAIR also enhances ligand-independent ER functions and participates in tamoxifen resistance [39]. The lncRNA H19 can modulate ER function as it has been shown to suppress the ER-activated Wnt signaling pathway in breast cancer cells [40].

## 3. LncRNAs and ER Function in Other Disorders

Investigations in an animal model (C57BLK/6J mice) of diabetes mellitus have shown the beneficial effects of quercetin as one of the most active materials in grape pomace extract. Diabetes was induced in these mouse models by a high fat diet. Subsequently, the mice were divided into study groups that received quercetin and a control group without any supplementation. Notably, the anti-diabetic effects of this compound have been exerted through activation of the ERα in a pathway which is dependent on the lncRNA suppressor of hepatic gluconeogenesis and lipogenesis (lncSHGL) [41]. The interaction between lncRNAs and ER is also implicated in the pathogenesis of liver steatosis. The expression of NEAT1 is down-regulated in HepG2 cells following knockdown of ERα. The 3′ terminal region of nuclear enriched abundant transcript 1 (NEAT1) has been shown to interact with ERα to enhance expression of Aquaporin 7 (AQP7) and subsequently inhibit liver steatosis. Thus, NEAT1 participates in the activation of ERα to modulate AQP7-associated hepatic steatosis [42]. Investigations in ovarian endometriotic tissues have shown the downregulation of the steroid receptor RNA activator 1 (SRA) lncRNA and ERα, but over-expression of SRA protein and ERβ in comparison with normal endometrial tissues. The silencing of SRA1 enhanced ERα levels but decreased ERβ levels in endometriotic stromal cells. Moreover, SRA1-silencing decreased proliferation and enhanced apoptosis in these cells. Thus, the SRA-mediated regulation of ER in ovarian endometriosis participates in the proliferation of endometriotic stromal cells [43].

Table 1 summarizes the results of studies which assessed the functional link between ERs and lncRNAs in human disorders.

## 4. miRNAs and ER Function in Cancers

Several ER-associated miRNAs have been dysregulated in cancer. In addition, miRNAs can participate in the anti-proliferative effects of therapeutic compounds. For instance, treatment of ERα-positive breast cancer cells with the O-methylated isoflavone biochanin A can enhance the expression of miR-375. This compound influences a feedback loop between miR-375 and ERα, thus enhancing the proliferation of breast cancer cells [51]. Moreover, low concentrations of formononetin have been shown to enhance the proliferation of ERα-positive cells human umbilical vein endothelial cells (HUVEC) and MCF-7. Formononetin increases the expression of ERα, miR-375, p-Akt, and Bcl-2. Thus, this agent induces the proliferation of ERα-positive cells via the induction of miR-375 expression through a miR-375/ERα feedback loop [52]. The ERα-miR-375-PTEN-ERK1/2-bcl-2 axis is also involved in the effects of this substance in the enhancement of the proliferation of CNE2 cells [53]. Curcumin has induced apoptosis in osteosarcoma cells through the induction of the miR-125a/ERRα signaling pathway [54]. Calycosin has been shown to stimulate apoptosis in colorectal cancer cells, by modulating the ERβ/miR-95 and IGF-1R, PI3K/axes [55]. On the other hand, myocardin suppresses the ERα-associated proliferation of MCF-7 cells by interfering with ER-associated transcriptional induction, mostly via the suppression of the ERα function. Notably, myocardin promotes the expression of miR-885, a miRNA that inhibits the translation of E2F1 [56].

A number of miRNAs have been shown to alter the expression of ER or other genes participating in the associated pathways. High-throughput sequencing of the transcriptome of the MCF-7 cell line showed increased expression of miR-335-5p and miR-335-3p. These miRNAs suppress the expression of genes participating in the ERα signaling pathway and confer tomoxifen-resistance in these cells. Therefore, although miR-335 has been regarded as a tumor suppressor miRNA, it exerts an oncogenic effect by enhancing agonistic estrogen signaling in the context of cancer [57]. The over-expression of miR-206, miR-133a, and miR-27b has suppressed the proliferation and migration of MCF-7 cells. Such tumor-suppressive effects of these miRNAs were exerted through the modulation of ERα and Aryl Hydrocarbon Receptor (AhR) signaling pathways. Thus, these miRNAs have been suggested as promising prognostic biomarkers and therapeutic targets in breast cancer [58]. Expression of miRNAs vary in association with the invasiveness of breast cancer cells. For instance, expressions of miR-221 and miR-222 in in situ tumors were inversely correlated with ER. However, in pure invasive breast cancer cells, there was a positive correlation between the expression of these miRNAs and *TIMP3* expression. Thus, the enhancement of expressions of miR-221/222 may be a critical occurrence in the development of in situ carcinomas. These miRNAs underscore the putative differences between different types of breast cancer [59]. The ERα-miR-1271-SNAI2 axis is involved in the regulation of transforming growth factor (TGF)-β-induced breast cancer progression [60]. miR-301a-3p and miR-129 have been shown to inhibit estrogen signaling through direct interaction with *ESR1* and *ESR2* genes, respectively [61,62]. In addition, miR-148a and miR-22 inhibit *ESR1* expression to inhibit the viability and migration of breast cancer cells [63,64]. miR-142-3p has also been identified as an inhibitor of ESR expression in ER-positive breast cancer cells [65]. The regulation of *ESR1* expression by miR-135b has implications in the pathogenesis of breast and prostate cancer [66]. On the other hand, decreased levels of miR-497 participate in the cell proliferation, migration, and invasion of ER-negative breast cancer by targeting ERα [67]. Conversely, miR-203 suppresses the estrogen-induced viability, migration and invasion of ERα-positive breast cancer cells [68]. miR-107-5p has a direct interaction with *ESR1* to enhance the proliferation and invasion of endometrial carcinomas [69]. On the other hand, miR-195 suppresses epithelial–mesenchymal transition by targeting G protein-coupled *ESR1* in endometrial carcinomas [70]. In hepatocellular carcinomas (HCCs), HBx protein-associated induction of miR-221 enhances the proliferation of cancer cells by targeting ERα [71]. miR-1280 has been implicated in the pathogenesis of thyroid carcinomas by targeting ERα [72]. miR-181a-5p has been identified as a mediator of ERβ-associated suppression of cholesterol production in triple-negative breast cancer [73]. ER-mediated miR-486-5p modulation of Olfactomedin 4 (OLFM4) is also implicated in ovarian cancer [74]. Importantly, miR-320a and miR-451a have been shown to sensitize tamoxifen-resistant breast cancer cells to this agent by targeting ERRγ and 14-3-3ζ, ERα, and autophagy, respectively [75,76]. Moreover, miR-27a and miR-125a-3p sensitize breast cancer cells to endocrine therapy [77,78]. On the other hand, miR-1271 participates in letrozole resistance through the inhibition of ERα expression [79]. Finally, the ERα/miR-124/AKT2 axis is involved in the pathogenesis of breast cancer [80].

Meanwhile, ERs have also been observed to affect the expression of miRNAs. For example, E2 has been shown to enhance the expression of miR-199a-3p in triple-negative breast cancer cells [81]. Moreover, E2 alters the expression of 114 miRNAs in HUVEC cells. These miRNAs have been enriched in cell death and survival, lipid metabolism and reproductive system function [82]. ERα-mediated upregulation of miR-590 alters the expression of FAM171A1, which contributes to breast cancer invasiveness [83]. Moreover, miR-27a/b and miR-494 participate in the estrogen-mediated down-regulation of the expression of tissue factor pathway inhibitor α [84]. Dehydroepiandrosterone induction of G-protein-coupled ER has enhanced miR-21 expression in human HCC [85]. ESRβ has been shown to enhance bladder cancer growth and invasion through the modulation of the miR-92a/DAB2IP axis [86]. The expression of miR-338-3p is modulated by estrogen through GPER in breast cancer cells [87].

ERβ has also been shown to modulate the expression of miRNAs. The downregulation of ERβ in MDA-MB-231 breast cancer cells significantly alters the miRNA profile, especially the levels of miR-10b, miR-200b and miR-145. Further experiments showed that the specific effect of estradiol on miRNA signature is determined by the ER status of breast cancer cells. miR-10b and miR-145 have been shown to modulate the EMT process and function of principal matrix molecules that participate in breast cancer aggressiveness. The inhibition of ERβ in MDA-MB-231 cells induces changes in cell behavior and in extracellular matrix proteins in association with the modulation of the miRNA profile. Interfering with the ERβ-modulated miR-10b and miR-145 has been suggested as a putative approach to detect and treat breast cancer [88]. Moreover, ERβhas been shown to decrease colon cancer metastasis via the miR-205-PROX1 axis [89].

The expression of miR-21 has been shown to be negatively correlated with the expression of ER [90], yet the details of the interaction between this miRNA and ER have not been elucidated.

miRNAs can regulate both the expression and activity of ER. For instance, miR-1, miR-9, miR-18a/b, miR-19a/b, miR-20a/b and miR-22 directly target the *ESR* gene to suppress its expression [91]. In addition, miRNAs can alter ER activity. For instance, miR-22 can also indirectly suppress ERα activity via the inhibition of the function of the ERα transcriptional coactivator, Sp1 [92]. Moreover, miR-206 has been predicted to target ERα coactivators Nuclear Receptor Coactivator 1 (NCOA1)/SRC1 and NCOA3/SRC3 [93].

## 5. miRNAs and ER Function in Other Disorders

A variety of studies have associated miRNA activity with ER function. For example, miR-320a has been found to be upregulated in the placental samples of preeclamptic patients and over-expression of this miR-320a inhibits trophoblast invasion [94]. This miRNA targets estrogen-related receptor gamma (ERRγ) and inhibits its transcription and translation. Thus, miR-320a upregulation leads to abnormal placentation by regulating ERRγ [94]. This miRNA has established roles in the proliferation, migration, invasion, and apoptosis of both trophoblasts and endothelial cells through modulation of this receptor [95]. Furthermore, the interactions between ER and miRNAs have been implicated in the pathogenesis of intervertebral disc degeneration (IDD). A comparison between normal and degenerated human cartilaginous endplates (CEP) tissues from individuals with idiopathic scoliosis and IDD, respectively, revealed lower levels of aggrecan, collagen II, TGF-β and ERα, but higher amounts of MMP-3, adamts-5, IL-1β, TNF-α, IL-6, and miR-221 in degenerated CEP tissues. E2 was shown to upregulate the expression of aggrecan and collagen II, and the production of TGF-β in degenerated CEP, thus protecting these cells from degeneration. Functional studies confirmed the direct interaction between ERα and miR-221. This miRNA may weaken the shielding influences of E2 in degenerated CEP cells via the inhibition of ERα [96]. miR-203-3p has been shown to be over-expressed in the nucleus pulposus (NP) tissues of high-grade IDD patients compared with low-grade IDD patients. Notably, the expression of this miRNA has been negatively correlated with ERα expression. miR-203-3p has been shown to directly target ERα in the NP cells of IDD patients [97]. The role of miR-203 in the regulation of ERα and cartilage degradation in IL-1β-stimulated chondrocytes has been verified in another study [98]. The suppression of miR-203 expression has improved osteoarthritis cartilage degradation in an animal model [99]. A number of miRNAs such as miR-26b-3p are involved in the regulation of osteoblast differentiation [100]. The expression of miR-92 has been inversely correlated with ERβ1 expression in the uterosacral ligaments of females with pelvic organ prolapse (POP) [101]. A number of dysregulated miRNAs in myasthenia gravis patients, namely miR-21-5p, let-7a, and let 7f, have been shown to have *ESR1*-binding sites [102] implying the role of ER signaling in this autoimmune disorder. ER-associated miRNAs are also involved in trophoblast invasion. The inhibition of miR-18a expression has suppressed the invasion and enhanced the apoptosis of human trophoblast cells by targeting the *ESR1* gene [103]. miR-18a also participates in the pathogenesis of coronary heart disease via targeting ER [104]. The suppression of miR-148a has been shown to protect against ovariectomy-associated osteoporosis by modulation of ERα [105]. The suppression of miR-181a protects against transient focal cerebral ischemia by targeting ERα in the astrocytes [106]. miR-26b-3p participates in the regulation of the proliferation of mesenchymal stem cells by targeting the ESR-CCND1 axis [107].

Table 2 and Table 3 summarize the functions of up- and downregulated ER-associated miRNAs in human disorders.

## 6. CircRNAs and ER Functions in Cancer

CircRNAs comprise a group of non-coding RNAs that are particularly stable and also have been associated with ER function. Similar to lncRNAs, they regulate the expression of several genes that are involved in diverse physiological and pathological processes. A possible mechanism for the participation of circRNAs in carcinogenesis is that they might serve as miRNA sponges [114]. Recently, investigators have assessed the circRNA signature in breast cancer and nearby normal tissues. The authors detected the differential expression of 1155 circRNAs between these two sets of samples. hsa_circ_103110, hsa_circ_104689 and hsa_circ_104821 were among the upregulated circRNAs, while expressions of hsa_circ_006054, hsa_circ_100219 and hsa_circ_406697 were decreased in breast cancer tissues [115]. Using a bioinformatics approach to recognize expressed circRNAs in breast cancer cell lines and clinical samples, greater quantities of circRNAs were detected in adjacent normal tissues of ER+ breast cancer samples compared to tumor samples [116]. Furthermore, an inverse correlation was observed between the quantities of circRNAs in adjacent normal tissues of ER+ samples and the risk-of-relapse proliferation score for proliferating genes. The latter finding implies that circRNA levels may be indicative of cell proliferation in breast cancer [116]. An examination of highly expressed circRNA isoforms spliced from *ESR1* that were differentially expressed in ER+ breast cancer tissues compared with other malignancies and normal breast tissue were found to be resistant to destruction by ribonuclease R, while the corresponding linear mRNA was prone to degradation by this enzyme. Based on the stability of circRNAs in the plasma, these circRNAs were proposed as putative biomarkers for the early non-invasive diagnosis of cancer [117]. In a recent study, the expression signature of circRNA was compared between ER-positive breast cancer tissues and their nearby non-tumor tissues. In total, more than 3000 differentially expressed circRNAs were identified between these two sets of samples. These circRNAs were enriched cancer-associated pathways. In addition, hsa_circ_0087378 has been verified to be downregulated in ER+ breast cancer samples and the hsa_circ_0087378-miR-1260b-SFRP1 axis has been recognized as an important regulatory axis in this type of cancer [118]. Finally, in ERα-associated circ-SMG1, the 72/miR-141-3p/gelsolin axis has been proposed as a strategy for the inhibition of HCC cell invasion [109]. Taken together, cicRNAs are potential regulators of ER signaling and their dysregulation participates in the pathogenesis of ER-related cancers, especially breast cancer. However, functional studies in this field are scarce.

## 7. Discussion

Several miRNAs and lncRNAs participate in the regulation of ER functions. Figure 2 depicts the mechanism of participation of a number of these transcripts in the regulation of ER in breast cancer. ER-associated ncRNAs regulate cancer initiation, progression and metastasis. Their role in the modulation of metastatic potential of cancer cells has been highlighted through the observed association between the expression of some ncRNAs such as miR-10b, miR-145 and miR-497 and the expression of MMP proteins. In addition, miRNAs might have diverse targets that modulate various steps in the carcinogenesis process. For instance, the ER-associated miRNA miR-21 has been shown to target several genes, namely *TIMP3*, *PDCD4*, *PTEN*, *TPM1* and *RECK*, which participate in numerous aspects of cancer progression including invasion, angiogenesis and metastasis [119]. The lncRNA DSCAM-AS1, which is expressed in ERα+ breast cancer tissues, is absent from pre-neoplastic samples [36], indicating a role for this lncRNA in the primary steps of breast carcinogenesis.

Understanding the molecular mechanisms of such contributions has practical significance in the management of several human disorders such as breast cancer. The observed endocrine resistance in a proportion of patients with this kind of cancer can be at least partly explained by the dysregulation in the expression of miRNAs/lncRNAs. Resistance to endocrine therapies is complicated process that involves epigenetic alterations in the *ESR1* gene, alternative splicing events, post-translational alterations and altered recruitment of co-regulators of ER; it also affects the tumor milieu and several other mechanisms [120]. Almost all aspects of this complex process can be modulated by non-coding RNAs. Numerous miRNAs directly bind with the *ESR1* gene to suppress its expression and induce resistance to estrogen/ERα-targeted therapeutic modalities in cancers [91]. Moreover, a number of herbal compounds have been shown to affect the functional links between ERs and non-coding RNAs, thus exerting anti-proliferative or proliferative effects on cancer cells.

Consistent with the widespread expression pattern of some types of ERs, the dysregulation of ER-associated lncRNAs/miRNAs has been involved in the pathogenesis of several disorders in almost all organ systems such as the reproductive, musculoskeletal, cardiac and gastrointestinal, endocrine systems, among others. The contribution of ER is not confined to the malignant status of these tissues. Myasthenia gravis, IDD, POP and preeclampsia represent a few examples of the significance of the functional link between non-coding RNAs and ERs. Based on the presence of complex interactions between miRNAs and ERs, and the competing effects of ERα and ERβ in some situations, a comprehensive assessment of miRNA signatures is needed to elaborate the underlying mechanisms of ER-associated pathologies. LncRNAs and miRNAs represent molecular biomarkers for some of these conditions. The therapeutic potential of these transcripts has also been assessed in some instances. For instance, the results of an animal study revealed that miR-181a silencing can inhibit transient focal cerebral ischemia by targeting ERα in astrocytes [106]. Moreover, an in vitro study showed the effects of the transfection of let-7 mimicked in human myofibroblasts on decreasing ERα expression and the related fibrotic pathways, thus highlighting this strategy for the treatment of idiopathic lung fibrosis [113]. Further assessments of the effects of the manipulation of non-coding RNAs expression in animal models would provide novel therapeutic strategies for ER-associated conditions.

## 8. Therapeutic Perspectives and Future Directions

LncRNAs and miRNAs are involved in several aspects of ER functions in both up- and downstream signaling pathways. Based on the crucial roles of ER-related signaling pathways in diverse human disorders, particularly cancers, the modulation of the expression of these transcripts can interfere with the progression of ER-related disorders or therapeutic responses to drugs that interfere with ER signaling. The most valuable results have been obtained from investigations in breast cancer, where the modulation of the expression of non-coding RNAs altered the response of cancer cells not only to tamoxifen, but also to herbal anti-cancer drugs. However, this field has been less explored for other malignant and non-malignant conditions that are associated with ER signaling. Thus, future in vitro and in vivo studies should assess the efficacy of non-coding RNA-mediated therapies in these conditions. Moreover, the interactive network between circRNAs and other non-coding RNAs, particularly miRNAs, is another research field which should be explored in future studies. This complex network might also alter the disease course or response to endocrine therapies. Considering the role of ER signaling in diverse aspects of the carcinogenesis process including both pathogenic events in cancer cells and the modulation of the tumor microenvironment, novel ER-targeted therapies are expected to effectively alter the course of cancer. Future investigations should focus on the design of the next generation of these molecules, considering their differential effects on diverse cell types, which are present in the tumor niche.

## Figures and Tables

**Figure 1 cancers-12-02162-f001:**
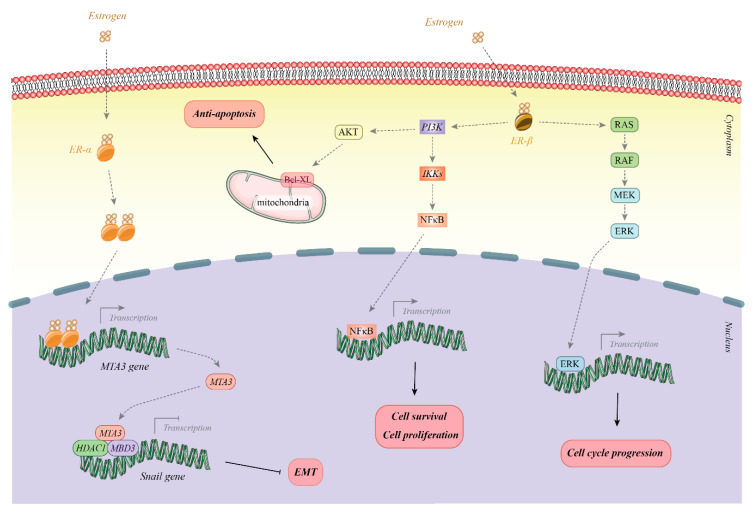
Overview of the estrogen receptor pathway and its key functions. Estrogen can bind with estrogen receptor (ERα or ERβ) to influence expression and activity of several signaling pathways that regulate cell cycle progression, cell survival and proliferation and epithelial–mesenchymal transition (EMT).

**Figure 2 cancers-12-02162-f002:**
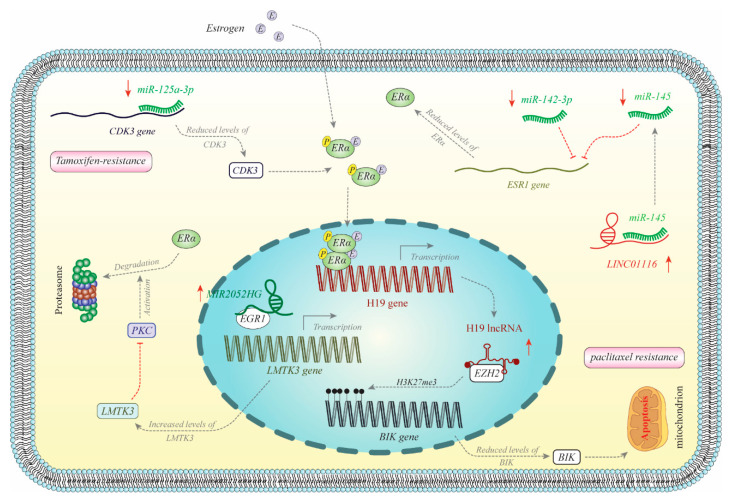
Schematic diagram illustrating mechanisms of noncoding RNA regulation of ER in breast cancer. Expression of miR-125a is increased in tamoxifen-resistant breast cancer cells. miR-125a binds to the 3′ UTR of CDK3 to inhibit its expression. CDK3 phosphorylates ER and increases its activity [78]. MIR2052HG is over-expressed in ERα-positive breast cancer cells. MIR2052HG facilitates binding of EGR1 with the promoter of the LMTK3 gene to enhance its expression. LMTK3 inhibits PKC and reduces ER degradation [45]. H19 is one the downstream targets of ERα. H19 is increased in ERα-positive breast cancer cells. H19 recruits EZH2 and enhances H3K27me3 of BIK promoter to reduce its expression, inhibit apoptosis and induce resistance to therapeutic options [30]. LINC01116 is increased in breast cancer tissues and acts as a sponge for miR-145, which binds with the 3′ UTR of *ESR1* to inhibit its expression. Thus, over-expression of LINC01116 enhances ER levels [29]. miR-142-3p is downregulated in ER-positive breast cancer cells. This miRNA binds with 3′ UTR of *ESR1* to inhibit its expression [65].

**Table 1 cancers-12-02162-t001:** List of dysregulated ER-associated lncRNAs and their functions in disorders.

Type of Disease	lncRNA	Expression Pattern	Human/Animal	Number of Clinical Samples	Assessed Cell Line	Targets/Regulators	Signaling Pathways	Function	Ref
Breast Cancer (BCa)	LINP1	Up	Mouse	-	MCF-7, T47D, TAMR	ER, GREB1, PGR	-	LINP1 by regulating ER expression and ER signaling promoted tamoxifen resistance in BCa cells.	[27]
	TMPO-AS1	Up	Human	ER+ samples (*n* = 32) ER- Samples (*n* = 83)	MCF-7, T47D, LTED, OHTR	*ESR1*, GREB1, WISP2	-	TMPO-AS1by regulating *ESR1*could promote hormone-refractory breast cancer progression.	[25]
	LINC01116	Up	Human	64 BCa tissues and 30 normal breast tissues, GSE54002 dataset	MCF-10A, MCF-7, HCC38, MDA-MB-231	miR-145, *ESR1*	-	LINC01116/miR-145/*ESR1* axis is involved in the pathogenesis of breast cancer.	[29]
	H19	Up	Human	-	MCF-7, ZR-75-1, MCF-7R, ZR-75-1R	BIK, EZH2	ERα	H19 expression was upregulated in paclitaxel (PTX)-resistant breast cancer cell lines. ERα/H19-/IK axis is involved in promoting chemoresistance in BCa.	[30]
	LASER1	Up	Human	primary BCa (*n* = 72) and late-stage relapse BCa patients (*n* = 24)	T47D	*ESR1*	-	LASERs via interacting with *ESR1* could promote late-stage relapse in breast cancer.	[44]
	MIR2052HG	Up	Human	TCGA dataset	-	ERα, LMTK3	PKC/MEK/ERK/RSK1	MIR2052HG via LMTK3 by recruiting EGR1 could regulate ERα level and endocrine resistance in breast cancer.	[45]
	LINC00707	Up	Human	TCGA dataset	MCF-10A, MCF-7, MCF-10AT	miR-206, ERα	-	Treatment with acetyl-11-keto-β-boswellic acid (AKBA, 25 µM) could inhibit breast lesion MCF-10AT cells by regulating LINC00707/miR-206 to reduce ERα.	[46]
	H19	Up	Human	-	MDA-MB231, MCF-7	-	ERβ, ERα	Treatment with E2 (10 nM) could regulate luminal progenitor cell differentiation through the ER/H19 axis.	[47]
	Glu	Down	Human	96 pairs of BCa and adjacent normal tissues	MDA-MB-468, HCC1806	VGLUT2	GPER, NMDARCaMK/ MEK-MAPK	Treatment with E2 (100 nM) could downregulate expression of Glu in MDA-MB-468 and HCC1806 cells. GPER-regulated lncRNA-Glu trough NMDAR-CaMK/MEK-MAPK pathway could promote invasion and metastasis in breast cancer.	[26]
	LINC00472	Down	Human	525 BCa patients	MCF-7, T47D, MDA-MB-231, Hs578T, SKBR3, ZR-75-1, 293T, H-6035	-	ERα, NF-κB	ERα/LINC00472 axis via suppressing NF-κB in breast cancer could inhibit BCa progression.	[48]
	RoR	Up	Human	TCGA dataset	MCF-7	-	ER, MAPK/ERK	E2-free treatment could upregulate the expression of RoR. RoR is involved in the pathogenesis of BCa via promoting MAPK/ERK and activating ER signaling pathways.	[49]
Papillary Thyroid Carcinoma (PTC)	H19	Up	Human	41 pairs of PTC and adjacent normal tissues	TPC-1, K-1, 293T	-	ERβ	Treatment with E2 could promote H19 transcription via ERβ in PTC cells. ERβ/H19 axis could promote cancer stem-like properties in PTC.	[31]
Liver steatosis	NEAT1	Up	Human	-	HepG2	AQP7	ERα	NEAT1 could promote steatosis via enhancement of ERα-mediated AQP7 expression in HepG2 cells.	[42]
Hepatocellular Carcinoma (HCC)	MTA1	Down	Mouse		HepG2, Hep3B, Hep3B-ERα	-	ERα	Treatment with E2 (100 nM) could downregulate the expression of MTA1 in Hep3B cells. ERα by inhibiting MTA1 could protect HCC from proliferation and metastasis	[35]
Type 2 Diabetes Mellitus (T2DM)	LncSHGL	Up	Mouse		HepG2, 293T, NCTC-149	ERα	-	Treatment with quercetin (100 mg/kg) via promoting lncSHGL through ERα could suppress the complications of T2DM.	[41]
Renal Cell Carcinoma (RCC)	HOTAIR	Up	Human	TCGA dataset	786-O, A498, ACHN, Caki-1	miR-138, miR-200c, miR-204, miR-217, ADAM9, CCND2, EZH2, VEGFA, VIM, ZEB1, ZEB2	ERβ	ERβ via regulating HOTAIR-miR-138/200c/204/217 associated ceRNA network could promote RCC progression.	[50]

**Table 2 cancers-12-02162-t002:** List of upregulated ER-associated miRNAs in disorders.

Diseases	microRNA	Human/Animal	Number of Clinical Samples	Assessed Cell Line	Targets/Regulators	Signaling Pathways	Function	Ref
Breast Cancer (BCa)	miR-375	Mouse	-	T47D, MCF-7	ERα, Bcl-2	-	Treatment with biochanin A (2–6 μM) could upregulate the expression of miR-375 in ERα+ BCa cells. Biochanin A by affecting a feedback loop of miR-375 and ERα could promote the proliferation of BCa cells.	[51]
	miR-375	Mouse	-	HUVEC, MCF-7	ERα, Bcl-2	Akt	Treatment with formononetin (2–6 mM) could increase the expression of miR-375. Therefore, formononetin by affecting a feedback loop of miR-375 and ERα could promote the proliferation of ERα-positive cell.	[52]
	miR-885	Human	-	MCF-7, MDA-MB-231	ERα, E2F1, PCNA	-	Myocardin via affecting miR-885 by ERα could inhibit ERα-mediated proliferation of breast cancer MCF-7.	[56]
	miR-27a/b, miR-494	Human	-	MCF-7, 293T	ERα, TFPIα	-	Treatment with 17a-ethinylestradiol (EE2, 100 nM) could upregulate the expression miR-27a/b and miR-494 through ERα in MCF7 cells. Then, mentioned-miRs via targeting TFPIα could be involved in the pathogenesis of BCa.	[84]
	miR-335-5p, miR-335-3p	Human	-	MCF-7, Hs578t MCF-7-TR, MCF-7-FR, MDA-MB-231, BT-549, MDA-MB-157,	ERα	-	miR-335 via inhibiting ERα could promote tamoxifen resistance in BCa cells.	[57]
	miR-221, miR-222	Human	Non-transformed breast tissue (*n* = 5), in situ carcinoma (*n* = 19), invasive BCa associated with non-invasive BCa (*n* = 12), pure invasive BCa (*n* = 27)	-	ER, *TIMP3*	-	miR-221/222 via ER/*TIMP3* axis could be involved in the pathogenesis of BCa.	[59]
	miR-10b	Human	-	MCF-7, MDA-MB-231	ERβ, MMP2, MMP7, MMP9, ECM, syndecan-1	ERK1/2	In estrogen-free medium, the expression of miR-10b was increased in MCF-7 cells. ERβ is as an epigenetic mediator of miR-10b to target EMT and ECM in mammary cancer.	[88]
	miR-206, miR-27b, miR-133a	Human	-	MCF-7	ERα, AhR	-	In the ERα silencing cell relative to scrambled, the expression of mentioned-miRs is increased. The forced increased expression of mentioned-miRs via targeting the ERα and AhR could suppress MCF-7 cell proliferation and migration.	[58]
	miR-301a-3p	Mouse and human	ER+/PR+/HER2+, ER−/PR−/HER2+, ER−/PR−/HER2−, and ER+/PR+/HER2− samples (*n* = 111)	ZR-751, MCF7, T47D, BT474, MCF10A	*ESR1*	-	miR-301a-3p upregulated in cancer stem-like cells. miR-301a-3p is able to suppress the ER signaling by directly inhibiting *ESR1* in the ER+ BCa cells in vitro and in vivo.	[61]
	miR-590-5p	Human cell line	-	MCF-7, T47D, MDA-MB-231, SKBR3, BT549, ZR-75, Hs-578T, HCC1937, MCF10A, SUM 149, SUM 159	FAM171A1	ERα	Loss of ERα/miR-590-5p axis via upregulating FAM171A1 resulted in the aggressiveness of triple-negative breast cancer (TNBC) cells.	[83]
	miR-21	Human	BCa samples (*n* = 75) including luminal A, luminal B(HER2+), luminal B(HER2−), basal-like type, and HER2 positive	-	ER, PR, HER2	-	miR-21 was upregulated in Basal-like and HER2 positive BCa types. miR-21/ER axis is involved in the pathogenesis of BCa.	[90]
	miR-206	Human	BCa samples (*n* = 75) including luminal A, luminal B(HER2+), luminal B(HER2−), basal-like type, and HER2 positive	-	ER, PR, HER2	-	miR-206 was upregulated in Luminal A and B types of BCa. miR-206/ER axis is involved in the pathogenesis of BCa.	[90]
	miR-199a-3p	Human cell line	-	HCC1806, HCC1937, MDA-MB-231, HMEC-184	GPER, YAP1, LATS1, E-cadherin, N-cadherin, Vimentin, VEGFA, AngII	Hippo	Treatment with G-1 G-1(1-[4-(-6-bromobenzol diodo-5-yl)-3a,4,5,9b tetrahidro3H5cyclopenta[c]quinolin-8yl]-ethanone) (1 μM, 48 h) and E2 (10 nM, 8 h) increased the expression of miR-199a-3p in MDA-MB-231 cells. The activation of GPER by regulating miR-199a-3p/CD151 axis could inhibit EMT process and proliferation, migration, and invasion of BCa cells.	[81]
	miR-181a-5p	Human	ERβ+ (*n* = 12) and ERβ− (*n* = 32) primary TNBC biopsies	HCC1806, MDA-MB-468, Hs 578T	ERβ	-	miR-181a-5p/ERβ axis is involved in suppressing of cholesterol biosynthesis in triple-negative breast cancer.	[73]
Preeclampsia	miR-320a	Human	Placentas from preeclampsia patient and healthy controls (*n* = 18/each)	HTR-8/SVneo	ERRγ	-	miR-320a by targeting ERRγ could inhibit trophoblast cell invasion.	[94]
Myasthenia Gravis (MG)	miR-21-5p	Human	MG patients without immunosuppression (*n* = 37), MG patients with immunosuppression (*n* = 14), normal controls (*n* = 27)	GM12878, H1-hESC, HeLa-S3, HepG2, HSMM, HUVEC, K562, NHEK, NHLF	ER, FOXO	NF-kB	miR-21-5p/ER axis is involved in the pathogenesis of MG.	[102]
Intervertebral disc degeneration (IDD)	miR-221	Human	Isolated human dysfunction of cartilaginous endplates (CEP) tissues (normal, *n* = 3) and (IDD patients, *n* = 15)	Normal and degenerated CEP cells	ERα	TGF-β	Treatment with estrogen (0.1–1 µM) could attenuate intervertebral disc degeneration by miR-221 through targeting ERα.	[96]
	miR-203-3p	Human	Nucleus pulposus samples were from IDD patients (*n* = 27) with different grades (G1-3)	Human nucleus pulposus	ERα	-	Suppression of miR-203-3p via upregulating ERα could inhibit lipopolysaccharide-induced human IDD and inflammation.	[97]
Endometrial carcinoma (EC)	miR-195	Human		AN3-CA, Hec1A	GPER	PI3K/AKT	miR-195 was overexpressed in the mimics group. Overexpression of miR-195 by targeting GPER could inhibit EMT process in EC.	[70]
EC	miR-107-5p	Human	EC samples (*n* = 71), normal controls (*n* = 26)	Ishikawa, 293T HEC-1B	ERα	-	miR-107-5p via targeting ERα could promote tumor proliferation and invasion in endometrial carcinoma.	[69]
Pelvic Organ Prolapse (POP)	miR-92	Human	56 POP patients and 48 non-POP control	-	ERβ1	-	miR-92 via downregulating ERβ1 in the cervical portion of uterosacral ligaments could be involved in the pathogenesis of POP.	[101]
Trophoblast Carcinoma	miR-18a	Human		JEG-3	ERα	-	Suppression of miR-18a expression by targeting ERα could promote apoptosis and inhibit invasion and of human trophoblast cells.	[103]
Hepatocellular Carcinoma (HCC)	miR-221	Human		HepG2, MCF-7, HepG2.215	ERα	-	Hepatitis B virus X protein (HBx) could increase the expression of miR-221 and miR-221 via suppressing ERα could promote HCC cancer cell proliferation.	[71]
HCC	miR-21	Human		HepG2, Hep3B, SK-HEP-1	GPER, *PDCD4*, ERα36	PI3K/AKT, EGFR, MAPK, ERK1/2	Treatment with DHEA (10 nM) via activating GPER could stimulate transcription of miR-21 in HCCs.	[85]
Osteoporosis	miR-148a	Rat		MC3T3-E1	ERα	PI3K/AKT	Downregulation of miR-148a via PI3K/AKT signaling by ERα could protect against ovariectomy-induced osteoporosis.	[105]
Nasopharyngeal Cancer	miR-375	Rat		CNE2	ERα, *PTEN*, Bcl-2	ERK1/2	Treatment with formononetin (0.3 µM) increased miR-375 levels in CNE2 cells. Low concentration of formononetin through the ERα/miR-375/*PTEN*/ERK1/2/Bcl-2 pathway could promote the proliferation of ER+ cells.	[53]
Osteoarthritis	miR-203		Osteoarthritis postmenopausal patients (*n* = 34), normal controls (*n* = 20)	Human chondrocytes	ERα	-	miR-203 via regulating ERα could enhance cartilage degradation in IL-1β-stimulated chondrocytes.	[98]
Osteoarthritis	miR-203	Rat		-	ERα	-	Inhibition of miR-203 by targeting ERα could ameliorate osteoarthritis cartilage degradation in the postmenopausal rat model.	[99]
Osteosarcoma	miR-125a	Human		U2OS, MG63	ERRα	-	Treatment with curcumin (20 mM) by suppressing ERRα through upregulation of miR-125a could promote osteosarcoma cell death.	[54]
Cerebral I/R Injury	miR-181a	Mouse		Primary astrocyte	ERα	-	Inhibition of miR-181a by targeting ERα could enhance E2 (20 nM)-mediated stroke protection in females.	[106]
Colorectal Cancer (CRC)	miR-129	Human	18 pairs of CRC tissue samples and adjacent normal tissues	HCT116	ERβ, PCNA, Caspase-3	-	The high expression level of miR-129 by targeting ERβ could contribute to aberrant CRC cell proliferation and migration.	[62]
Coronary Heart Disease (CHD)	miR-18a	Human	A total of 120 blood samples collected from CHD patients and normal controls	HUVECs	ER	PI3K/Akt/mTOR	miR-18a via the ER/PI3K/Akt/mTOR axis could regulate CHD development.	[104]
Bladder Cancer	miR-92a	Mouse, TCGA database		UMUC3, J82	DAB2IP	ERβ	ERβ via alteration of miR-92a/DAB2IP signals could promote bladder cancer growth and invasion.	[86]
-	miR-30b-5p, miR-487a-5p, miR-4710, miR-501-3p	Human		HUVEC	ERα, ERβ, GPER	-	Treatment with E2 (1 nmol/L) increased the expression of mentioned-microRNAs in HUVEC cells. In human endothelial cells, expressed miRNA pathways linked to E2 through ER demonstrate that estrogen can modulate endothelial function.	[82]
-	miR-26b-3p	Human		hUC-MSC	*ESR1*, CCND1	-	miR-26b-3p by targeting *ESR1* could regulate the proliferation of human umbilical cord-derived mesenchymal stem cells.	[107]

**Table 3 cancers-12-02162-t003:** List of downregulated ER-associated miRNAs in disorders.

Diseases	microRNA	Human/Animal	Number of Clinical Samples	Assessed Cell Line	Targets/Regulators	Signaling Pathways	Function	Ref
Breast Cancer (BCa), Prostate Cancer (PC)	miR-135b	Human	MicMa cohort (101 primary breast carcinoma samples), METABRIC database, PCa samples (*n* = 47)	HMEC, DU-145, PNT2, MCF-7, PC-3 LNCap, 22Rv1, BT-474, JIMT-1, KPL-4, VCaP	ERα, HIF1AN	HIF1α	miR-135b expression is downregulated in the ERα+ tumors relative to the ERα- tumors. miR-135b could affect breast and prostate cancer cell growth by regulating ERα, AR and HIF1AN and regulate proliferation in ERα+ BCa and AR+ PCa cells.	[66]
BCa	miR-320a	Human cell line		T47D, MCF-7	ERR*γ*, ARPP-19, c-Myc, Cyclin D1	-	Knockdown of miR-320a by targeting ARPP-19 and ERR*γ* could reduce the sensitivity of tamoxifen in ER+ breast cancer cell lines.	[75]
	miR-338-3p	Human cell line		SkBr3, CAFs	GPER	-	Treatment with E2 (100 nM) decreased the expression of miR-338-3p in SkBr3 cancer cells and cancer-associated fibroblasts (CAFs). In the regulation of miR-338-3p by E2, the GPER is involved.	[87]
	miR-451	Human cell line		MCF-7, MCF-7/Dox	ER, ABCB1, Caspase-3, Pgp	-	Resistance to doxorubicin is correlated with dysregulation of the miR-451/ER axis in BCa cell line.	[108]
	miR-451a	Human cell line		MCF-7, LCC2	ERα 14-3-3ζ	AKT, mTOR	The expression of miR-451a is decreased in LCC2 cells compared to MCF-7 cells. Treatment with tamoxifen (TAM) could downregulate ERα expression and upregulate 14-3-3ζ expression. Hence, overexpression of miR-451 via activating ERα and inactivating 14-3-3ζ could enhance the sensitivity of breast cancer cells to TAM.	[76]
	miR-497	Human	ERα+BCa (EPBC, *n* = 30 tissues), ERα−Bca (ENBC, *n* = 30 tissues)	MCF-7, T47D, MDA-MB-231, SKBR3	ERRα, MIF, MMP9		The expression of miR-497 is reduced in ENBC tissues. The downregulation of miR-497 by targeting ERRα could contribute to proliferation, migration, and invasion of ERα– BCa cells.	[67]
	miR-124	Human	46 pairs of BCa samples and adjacent normal tissues), ERα− (*n* = 17), ERα+ (*n* = 29)	MCF-7, MDA-MB-231	AKT2	ER	Treatment with 17β-estradiol (E2, 10 nM) decreased miR-124 levels in MSF-7 cells through ER. However, overexpression of miR-124 by targeting AKT2 suppressed tumor growth and angiogenesis in MCF7 cells. Therefore, the ERα/miR-124/AKT2 axis is involved in BCa development.	[80]
	miR-27a	Human cell line		MCF-7, T47D	ERα	-	In tamoxifen (20 µm)-resistant cells, the expression of ERα and miR-27a decreased. miR-27a based on a positive feedback loop with ERα could sensitize luminal A BCa cells to selective estrogen receptor modulators (SERMs) treatments	[77]
	miR-203	Human	22 pairs of BCa tissue samples and adjacent normal tissues	MCF-7	ERα	-	Treatment with E2 (10 mM) decreased miR-203 levels in MCF-7 cells. miR-203 via directly suppressing ERα could inhibit the viability, migration, and invasion of estrogen-dependent BCa cells.	[68]
	miR-148a	Human cell line		MCF-7, 293T	ERα	-	Overexpression of miR-148a via inhibiting ERα could suppress the E2 (1 mM)-induced viability and migration of ERα+ BCa cells.	[63]
	miR-1271	Mouse, TCGA database		MCF-7, T47D, BT474, BT549, MDA-MB-468, MDA-MB-231	SNAI2	TGF-β	The ERα-miR-1271-SNAI2 feedback loop is involved in the regulation of TGF-β signaling during BCa progression and development.	[60]
	miR-142-3p	Human	20 pairs of primary BCa and adjacent normal breast tissues	MDA-MB-231, MCF-7, 293T	*ESR1*	-	miR-142-3p via inhibiting estrogen *ESR1* could act as a tumor suppressor in ER+ BCa.	[65]
	miR-22	Human	50 pairs of primary BCa and adjacent normal breast tissues	MDA-MB-231, MCF-7, T47D, SKBR-3, HBL-100	ERα, NK1R-Tr	ERK1/2	miR-22 via targeting ERα and NK1R-Tr could inhibit proliferation, invasion, and metastasis of BCa cells.	[64]
	miR-1271	Mouse and human	non-responding tumor tissues (*n* = 30) and responding tumor tissues (*n* = 40)	MCF-7, BT474, BT483, HCC1007, HCC1569, HCC1187, MDA-MB-231, MDA-MB-361, MDA-MB-415	ERα, DDIT3	MAPK	miR-1271 was downregulated in letrozole-resistant BCa tissues/cells. miR-1271 by inhibiting ERα could alter letrozole resistance in BCa.	[79]
	miR-145	Human cell line	-	MCF-7, MDA-MB-231	ERβ, MMP2, MMP7, MMP9, ECM, syndecan-1	ERK1/2	In an estrogen-free medium, the expression of miR-145 was decreased in MCF-7 cells. ERβ is as an epigenetic mediator of miR-145 to target EMT and ECM in mammary cancer.	[88]
	miR-125a-3p	Human	37 cancerous tissues paired with noncancerous samples	MCF-7, MDA-MB-435, MDA-MB-231	ERα, CDK3	-	miR-125a-3p via inhibiting ERα transactivation and targeting CDK3 could override tamoxifen resistance in ER+ breast cancer.	[78]
Thyroid Cancer (TC)	miR-1280	Human	12 pairs of follicular thyroid cancer and adjacent non-neoplastic tissues	FTC133, TT, Nthy-ori 3-1	ERα	ERK	Overexpression of miR-1280 by inhibiting ERα could promote cell proliferation and invasion in TC cells.	[72]
Ovarian Cancer (OC)	miR-486-5p	Human	Ovarian serous adenocarcinoma (*n* = 6) and normal ovary (*n* = 8)	SKOV3, HO8910-pm	OLFM4	ER	Treatment with 17β-estradiol (E2) decreased miR-486-5p levels in SKOV3 cells. However, ER/miR-486-5p/OLFM4 axis could be involved in the development and progression of ovarian cancer.	[74]
Hepatocellular Carcinoma (HCC)	miR-141-3p	Mouse and human	24 pairs of HCC and adjacent normal liver tissues	SK-HEP-1, 293T, HA22T	ERα, circRNA-SMG1.72, gelsolin	-	ERα via altering the ERα/circRNA-SMG1.72/miR-141-3p/GSN axis could suppress HCC cell invasion.	[109]
Prostate Cancer (PC)	miR-135a	Human cell line	-	MDA-MB-231, PC3, LNCaP	ERRα	-	Overexpression of miR-135a via suppressing ERRα could inhibit the invasion of prostatic cancer cells.	[110]
Osteoporosis	miR-210-3p	Human cell line	-	rBMSCs	-	Wnt	In ERα-deficient model, miR-210-3p through Wnt signaling could promote adipogenic differentiation and inhibit osteogenic differentiation of rBMSCs.	[111]
Colorectal Cancer (CRC)	miR-95	Human cell line		SW480, LoVo, HeLa	ERβ	IGF-1R, PI3K/Akt	Calycosin (80 μM) could inhibit human CRC cell proliferation by targeting miR-95 through ERβ.	[55]
CRC	miR-205	Human	RNA-seq data (233 primary colon cancer specimens and 21 adjacent normal tissues)	SW480, HT29, HCT116, SW403, SW620, 293T	PROX1	ERβ	ERβ upregulates miR-205 and that miR-205 by targeting and repressing PROX1 could reduce the proliferative and metastatic potential of the CRC cells.	[89]
-	miR-320a	Human cell line	-	HTR.8/Svneo, HUVECs	ERRγ, VEGF, Ang-1, HCG, HSD3B1	-	Overexpression of miR-320a by inhibiting ERRγ could downregulate the expression of trophoblast-associated markers and angiogenesis-related factors in trophoblasts and endothelial cells.	[95]
-	miR-26b-3p	Human cell line	-	MC3T3-E1, 293T	ERα	-	miR-26b-3p via targeting ERα could regulate osteoblast differentiations.	[100]
-	miR-378h, miR-1244	Human cell line	-	HUVEC	ERα, ERβ, GPER	-	Treatment with E2 (1 nmol/L) decreased the expression of mentioned-microRNAs in HUVEC cells. In human endothelial cells, expressed miRNA pathways linked to E2 through ER demonstrate that estrogen can modulate endothelial function.	[82]
-	miR-221	Rat	-	-	ERα	ERK1/2, NF-κB, AKT	Treatment with E2 (40 μg/kg) by suppressing hepatic iNOS through the activation of the ERα could inhibition of ERK1/2-mir-221 axis.	[112]
Idiopathic Pulmonary Fibrosis (IPF)	let-7a/d	Mouse and human	lung tissue of IPF patients (*n* = 8) and normal controls (*n* = 6)	Myofibroblast	ERα, ERβ	TGF-β, AKT, SMAD	let-7/ER axis is involved in the pathogenesis of predominant pulmonary fibrosis.	[113]

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
