# Peer review of "Perspectives on the Role of Non-Coding RNAs in the Regulation of Expression and Function of the Estrogen Receptor"

_cancers, 2020, doi:10.3390/cancers12082162_

Round 1

Reviewer 1 Report

In this small review entitled "Non-coding RNAs Regulate Expression and Function of Estrogen Receptor", Taheri et al. elegantly present a compendium of non-coding RNAs and Estrogen Receptors in cancers. Albeit being a short review, the authors compiled a number of non-coding RNAs, including miRNAs and lncRNAs, and their roles in controlling the expression levels of ERs in cancers. The presentation via easy-to-read tables and a figure certainly qualifies this as a decent review. Below are a few comments/suggestions the authors may want to incorporate for substantially improving their review.

General major (essential) comments:

1. For me, the title sounds like a research article rather than a review. The authors may want to consider a minor revision for their title. Words such as "Perspectives on non-coding RNAs...." or "On the role of non-coding RNAs in regulating the function...in cancers"... etc. will be better.

2. A schematic that depicts the general functions and/or the estrogen receptor pathway will be easier for the readers to follow the introduction section in a better way. Ideally, this schematic may be a general one, without already portraying the miRNAs or any other perturbations known in literature. 

3. The tables are pretty well designed but I find the "Human/Animal" column very inconsistent. The authors write Mouse if it is mouse, but describe sample size when it comes to human samples. Please write Mouse or Human only in that column. If any, have an additional column describing the sample size and/or source of the datasets.

4. I find that there are several other studies which the authors have neglected to include in their review. Importantly, I did not find the mention of MIAT lncRNA, which is regulated in breast cancers (Li et al., BBRC, 2018; Alipoor et al., J Cellular Biochemistry, 2018). Alipoor et al, also show certain miRNAs regulated in breast cancers, which have to be included in this review.

5. In addition to the aforementioned missing articles, the authors should do a thorough data mining to include additional key references, those that also describe the high throughput identification of several lncRNAs. Example papers that ought to be included are:

Miano et al., Oncotarget, 2016; Wang et al., Scientific Reports, 2018; Wu et al., Scientific Reports, 2016; Mozdarani et al., Journal of Translational Medicine, 2020 (This paper describes the role of HOTAIR in estrogen and breast cancer context); Lin et al., Reproductive Sciences, 2016 (This paper describes SRA lncRNA, which is missing in this review!!!); Aiello et al., Scientific Reports, 2016 (This paper describes MALAT1 in prostate cancer); Ntini et al., Nat Communications, 2018. And there are several other papers that the authors need to dig in and include.

6. Certain other important reviews relevant to the current review (e.g., Schmitt and Chang, Cancer Cell, 2016) must be cited.

7. A short section on the therapeutic perspectives and future directions based on the available knowledge of non-coding RNAs and its link to ER and cancers will be valuable for this review. Something like "Where do we o from here...?"

In summary, this is an elegant review but the authors must take care of the above mentioned papers (and perhaps other very important ones relevant to this review) that must be included in this review.

Author Response

In this small review entitled "Non-coding RNAs Regulate Expression and Function of Estrogen Receptor", Taheri et al. elegantly present a compendium of non-coding RNAs and Estrogen Receptors in cancers. Albeit being a short review, the authors compiled a number of non-coding RNAs, including miRNAs and lncRNAs, and their roles in controlling the expression levels of ERs in cancers. The presentation via easy-to-read tables and a figure certainly qualifies this as a decent review. Below are a few comments/suggestions the authors may want to incorporate for substantially improving their review.

General major (essential) comments:

  1. For me, the title sounds like a research article rather than a review. The authors may want to consider a minor revision for their title. Words such as "Perspectives on non-coding RNAs...." or "On the role of non-coding RNAs in regulating the function...in cancers"... etc. will be better.

Response: We edited the title.

  1. A schematic that depicts the general functions and/or the estrogen receptor pathway will be easier for the readers to follow the introduction section in a better way. Ideally, this schematic may be a general one, without already portraying the miRNAs or any other perturbations known in literature.

Response: We added this picture.

  1. The tables are pretty well designed but I find the "Human/Animal" column very inconsistent. The authors write Mouse if it is mouse, but describe sample size when it comes to human samples. Please write Mouse or Human only in that column. If any, have an additional column describing the sample size and/or source of the datasets.

Response: We edited the tables accordingly.

  1. I find that there are several other studies which the authors have neglected to include in their review. Importantly, I did not find the mention of MIAT lncRNA, which is regulated in breast cancers (Li et al., BBRC, 2018; Alipoor et al., J Cellular Biochemistry, 2018). Alipoor et al, also show certain miRNAs regulated in breast cancers, which have to be included in this review. 2

Response: We added these references.

  1. In addition to the aforementioned missing articles, the authors should do a thorough data mining to include additional key references, those that also describe the high throughput identification of several lncRNAs. Example papers that ought to be included are:

Miano et al., Oncotarget, 2016; Wang et al., Scientific Reports, 2018; Wu et al., Scientific Reports, 2016; Mozdarani et al., Journal of Translational Medicine, 2020 (This paper describes the role of HOTAIR in estrogen and breast cancer context); Lin et al., Reproductive Sciences, 2016 (This paper describes SRA lncRNA, which is missing in this review!!!); Aiello et al., Scientific Reports, 2016 (This paper describes MALAT1 in prostate cancer); Ntini et al., Nat Communications, 2018. And there are several other papers that the authors need to dig in and include.

Response: We added these references.

  1. Certain other important reviews relevant to the current review (e.g., Schmitt and Chang, Cancer Cell, 2016) must be cited.

Response: We cited this reference.

  1. A short section on the therapeutic perspectives and future directions based on the available knowledge of non-coding RNAs and its link to ER and cancers will be valuable for this review. Something like "Where do we o from here...?"

Response: We added this section.

In summary, this is an elegant review but the authors must take care of the above mentioned papers (and perhaps other very important ones relevant to this review) that must be included in this review.

Reviewer 2 Report

In the article, the authors describe the role of non-coding RNAs  in estrogen signaling. The review is interesting. Nevertheless, is should be better organized and the selected information should be added to the article.

  • In the title, the authors declared to summarize to role of non-coding RNAs in regulation of expression and function of estrogen receptor. Nevertheless, they describe only the role of long non-coding RNAs and microRNAs. Is there anything known about the role of the other species of non-coding RNAs (e.g. circular RNAs) in estrogen signaling? If yes, it should be added to the article.
  • Short general introduction about different species of non-coding RNAs and their roles should be added to the article.
  • The text of the article should be organized in separate paragraphs, when the authors describe different lncRNAs, not as a single paragraph.
  • The sections about microRNAs should be better described in the text. miRNAs directly targeting ERs should be clearly distinguished in the text, as well as miRNAs directly regulated by ERs.
  • Misspellings should be corrected, eg. Line 48 “fututre”.

Author Response

In the article, the authors describe the role of non-coding RNAs  in estrogen signaling. The review is interesting. Nevertheless, is should be better organized and the selected information should be added to the article.

In the title, the authors declared to summarize to role of non-coding RNAs in regulation of expression and function of estrogen receptor. Nevertheless, they describe only the role of long non-coding RNAs and microRNAs. Is there anything known about the role of the other species of non-coding RNAs (e.g. circular RNAs) in estrogen signaling? If yes, it should be added to the article.

Response: We added a paragraph regarding the association between circRNAs and ER.

Short general introduction about different species of non-coding RNAs and their roles should be added to the article.

Response: We added such general introduction.

The text of the article should be organized in separate paragraphs, when the authors describe different lncRNAs, not as a single paragraph.

Response: We divided the text to distinct paragraphs.

The sections about microRNAs should be better described in the text. miRNAs directly targeting ERs should be clearly distinguished in the text, as well as miRNAs directly regulated by ERs.

Response: We have applied this comment.

Misspellings should be corrected, eg. Line 48 “fututre”.

Response: We corrected misspellings.